# MODERN HOPFIELD NETWORKS CANNOT SOLVE $\text{NC}^1$-HARD PROBLEMS

## ABSTRACT

Modern Hopfield networks (MHNs) have emerged as powerful tools in deep learning, capable of replacing components such as pooling layers, LSTMs, and attention mechanisms. Recent advancements have enhanced their storage capacity, retrieval speed, and error rates. However, the fundamental limits of their computational expressiveness remain unexplored. Understanding the expressive power of MHNs is crucial for optimizing their integration into deep learning architectures. In this work, we establish rigorous theoretical bounds on the computational capabilities of MHNs using circuit complexity theory. Our key contribution is that we show that MHNs are DLOGTIME-uniform $\text{TC}^0$. Hence, unless $\text{TC}^0 = \text{NC}^1$, a $\text{poly}(n)$-precision modern Hopfield networks with a constant number of layers and $O(n)$ hidden dimension cannot solve $\text{NC}^1$-hard problems such as the undirected graph connectivity problem and the tree isomorphism problem. We also extended our results to Kernelized Hopfield Networks. These results demonstrate the limitation in the expressive power of the modern Hopfield networks.

## 1 INTRODUCTION

Hopfield networks (Hopfield, 1982), initially introduced as associative memories capable of storing and retrieving patterns, have undergone significant advancements in recent years. These developments led to the emergence of modern Hopfield networks (Ramsauer et al., 2021). Modern Hopfield networks (MHNs) address the limitations of their predecessors, particularly in terms of storage capacity, retrieval speed, and error rates. An important feature of MHNs is their ability to function as specialized components within deep networks, integrating memory capabilities and supporting diverse functionalities. For instance, MHNs can replace conventional layers such as pooling layers, permutation-equivariant layers (Guttenberg et al., 2016; Ravanbakhsh et al., 2016), GRU (Cho et al., 2014), LSTM (Hochreiter, 1991; Hochreiter & Schmidhuber, 1997), and attention mechanisms (Vaswani et al., 2023; Bahdanau et al., 2016). Their energy-based formulation allows for rapid and efficient pattern retrieval, making them a compelling alternative to traditional memory structures in deep learning.

Understanding modern Hopfield networks' computational capabilities and limitations is critical for their effective application. Specifically, it is crucial to explore the computational operations these networks can implement and the complexity of the problems they can solve collectively. These investigations are vital not only for theoretical insights but also for guiding the development of more efficient and powerful models.

While prior research has primarily focused on the dynamics and capacity of modern Hopfield networks (Hu et al., 2024c; Wu et al., 2024a), their expressiveness from a circuit complexity perspective remains underexplored. This gap raises an important question:

*What are the fundamental limits of modern Hopfield networks in terms of circuit complexity?*

To address these questions, we turn to the framework of circuit complexity theory, which provides a robust method for analyzing the computational resources required to perform specific tasks. By mapping modern Hopfield networks to computational circuits, we can rigorously assess their capabilities and establish upper and lower bounds on the classes of problems they can solve.

In this work, we present a comprehensive theoretical investigation into the circuit complexity bounds of MHNs. Our approach involves analyzing the architecture of MHNs and the computational complexity of its components, such as the Hopfield layer. Hence, we show that uniform $\mathsf{TC}^0$ circuits can efficiently simulate these models.

Our main contributions can be outlined as follows:

- We prove that any $\mathrm{poly}(n)$-precision modern Hopfield networks or kernelized Hopfield networks with constant-depth and $O(n)$ hidden dimension is in DLOGTIME-uniform $\mathsf{TC}^0$ circuit family (Theorem 4.6 and 5.4).
- We prove that unless $\mathsf{TC}^0 = \mathsf{NC}^1$, a $\mathrm{poly}(n)$-precision modern Hopfield networks or kernelized Hopfield networks with constant layers, $O(n)$ hidden dimension cannot solve $\mathsf{NC}^1$-hard problems such as the undirected graph connectivity problems and tree isomorphism problems (Theorems 6.1, 6.2, 6.3 and 6.4).

**Roadmap.** In Section 2, we provide an overview of the related works. In Section 3, we introduce the notations and definitions needed for modern Hopfield networks. In Section 4, we analyze the circuit complexity for modern Hopfield networks. In Section 5, we focus on the circuit complexity results of kernelized Hopfield networks. In Section 6, we discuss the hardness results of the modern Hopfield networks. Finally, in Section 7, we summarize our theoretical results in the conclusion. We defer more preliminaries and technique details in Appendix.

## 2 RELATED WORKS

In this section, we introduce the related works of the Hopfield models and the complexity of Transformers. In Appendix A, we introduce more related works.

**Modern Hopfield Networks and Applications in Deep Learning.** Classical Hopfield networks (Hopfield, 1982; 1984; Krotov & Hopfield, 2016; 2018; 2020) have long been recognized for their ability to emulate human brain associative memory, primarily focusing on the storage and retrieval of memory patterns. Originally limited by linear memory capacity (Krotov & Hopfield, 2016), these networks now achieve exponential storage capabilities (Demircigil et al., 2017) and kernelized (Wu et al., 2024a), allowing for efficient handling of vast memory patterns. Architectural innovations (Hoover et al., 2023; Seidl et al., 2022; Fürst et al., 2022) have integrated Hopfield networks into models like Transformers, serving as advanced attention mechanisms (Vaswani et al., 2023). Their biological plausibility (Kozachkov et al., 2023; Krotov & Hopfield, 2020) has been a key focus, aligning computational models more closely with human associative memory. These developments have expanded applications across various fields, including tabular data learning (Xu et al., 2024), drug discovery (Schimunek et al., 2023), immunology (Widrich et al., 2020), time series forecasting (Wu et al., 2024a), reinforcement learning (Paischer et al., 2023), and large foundation models (Fürst et al., 2022), demonstrating their versatility in addressing complex problems.

## 3 PRELIMINARIES

In Section 3.1, we define useful notations for defining modern Hopfield networks. In Section 3.3, we introduce some components of modern Hopfield networks. In Section 3.4, we state the basic definitions and architectures of kernelized Hopfiled networks. The background knowledge about circuit complexity is introduced in Appendix C.

### 3.1 NOTATIONS

We define $\mathbb{N} := \{0, 1, 2, \ldots\}$ as the set of natural numbers and $\mathbb{R}$ as the set of real numbers. For any positive integer $n$, $[n]$ represents the set $\{1, 2, \ldots, n\}$. The vector $\mathbf{1}_n$ denotes an $n$-dimensional vector where all entries are ones. Given a matrix $A \in \mathbb{R}^{m \times n}$, $A_{i,j}$ refers to the element in the $i$-th row and $j$-th column, while $A^\top$ represents the transpose of $A$. The set $\{0, 1\}^*$ represents all binary strings of finite length. Specifically, for $x_i \in \{0, 1\}^*$, $x_i$ represents a finite binary string, with each bit being either 0 or 1. A language $L$ is characterized as a subset of $\{0, 1\}^*$.

## 3.2 FLOATING-POINT NUMBERS

This section presents key definitions necessary for our computational framework. We introduce basic concepts related to floating-point numbers and the definition of the operations, which are fundamental to implementing efficient Hopfield network computations.

**Definition 3.1** (Floating-point number, Definition 9 in (Chiang, 2024)). *A $p$-bit floating-point number is represented as a pair $\langle m, e \rangle$, where $m$ is the significand and $e$ is the exponent. Specifically, $m \in (-2^p, -2^{p-1}] \cup \{0\} \cup [2^{p-1}, 2^p)$ and $e \in [-2^p, 2^p)$. The floating-point value of $\langle m, e \rangle$ is the real number $m \cdot 2^e$. The set of all $p$-bit floating-point numbers is denoted as $\mathbb{F}_p$.*

To work with floating-point numbers effectively, we define specific rules for rounding and performing arithmetic operations:

**Definition 3.2** (Rounding, Definition 9 in (Chiang, 2024)). *For any real number or floating-point value $x$, we define $\text{round}_p(x)$ as the $p$-bit floating-point number with an even significand closest to $x$.*

Based on these definitions, we outline the main arithmetic operations for floating-point numbers required for Hopfield networks:

These operations are not merely theoretical, they can be practically implemented in hardware, as demonstrated by the following lemma:

**Lemma 3.3** (Basic floating-point operations in $\mathsf{TC}^0$, Lemmas 10, 11 in (Chiang, 2024)). *If the precision $p \leq \text{poly}(n)$, then the following results are valid:*

**Part 1.** *Let $x_1, x_2$ be two $p$-bits floating point numbers. The arithmetic operations of addition, multiplication, division, and comparison of $x_1$ and $x_2$, as described in (Chiang, 2024), can be executed using a DLOGTIME-uniform threshold circuit characterized by constant depth, and a size bounded by $\text{poly}(n)$. The maximum depth of the circuit required is denoted by $d_{\text{std}}$.*

**Part 2.** *Let $x_1, \ldots, x_n$ be $n$ $p$-bit floating-point number. Iterative multiplication of $x_1, \ldots, x_n$ can be executed with a DLOGTIME-uniform threshold circuit characterized by constant depth and a size bounded by $\text{poly}(n)$. The circuit depth for this operation is indicated as $d_{\otimes}$.*

**Part 3.** *Let $x_1, \ldots, x_n$ be $n$ $p$-bit floating-point number. Iterative addition of $x_1, \ldots, x_n$, where rounding is applied after the summation is completed, can be achieved using a DLOGTIME-uniform threshold circuit characterized by constant depth, and a size bounded by $\text{poly}(n)$. The depth required for this operation is represented by $d_{\oplus}$.*

This lemma shows that basic arithmetic operations can be efficiently executed in $\mathsf{TC}^0$ when operating on floating-point numbers with precision $p \leq \text{poly}(n)$.

**Lemma 3.4** (Computing $\exp$ in $\mathsf{TC}^0$, Lemma 12 in (Chiang, 2024)). *Assuming that precision $p \leq \text{poly}(n)$. For any $p$-bit floating-point number $x$, there exists a threshold circuit characterized by uniformity, constant depth, and a size bounded by $\text{poly}(n)$ that approximates $\exp(x)$, which has a relative error at most $2^{-p}$. The depth of the circuit required for this computation is denoted by $d_{\exp}$.*

**Lemma 3.5** (Approximating square root in $\mathsf{TC}^0$, Lemma 12 in (Chiang, 2024)). *Assuming that precision $p \leq \text{poly}(n)$. There exists a threshold circuit characterized by uniformity, constant depth, and a size bounded by $\text{poly}(n)$ that computes $\sqrt{x}$ for any $p$-bit floating-point number $x$, which has a relative error at most $2^{-p}$. The circuit depth required for computing $\sqrt{x}$ is denoted by $d_{\text{sqrt}}$.*

### 3.3 MODERN HOPFIELD NETWORK

With the above notations established, we proceed to introduce the foundational concepts of Modern Hopfield Networks.

The input query pattern is denoted as $x \in \mathbb{R}^d$, while memory patterns are represented by the matrix $\Xi = [\xi_1, \ldots, \xi_M] \in \mathbb{R}^{d \times M}$. Hopfield models are associative memory models based on energy functions, where the stored memory patterns $\Xi$ correspond to the local minima of these energy landscapes. For a given input query $x$, the model retrieves the most similar memory pattern by employing energy minimization algorithms, referred to as retrieval dynamics $\mathcal{T}$, initialized at $x$.

In (Ramsauer et al., 2021), the Modern Hopfield Model is introduced with a specific energy function $E$ and retrieval dynamics $\mathcal{T}$, which are further incorporated into deep learning frameworks due to their relation to transformer attention mechanisms (Vaswani et al., 2023). This approach enhances performance and provides a theoretical guarantee of exponential memory capacity. The energy function is defined as: $E(x) = -\text{lse}(\beta, \Xi^\top x) + \frac{1}{2}\langle x, x \rangle$, where the retrieval dynamics is given by

$$x^{\text{new}} = \mathcal{T}_{\text{Dense}}(x) = \Xi \cdot \text{Softmax}(\beta \Xi^\top x). \tag{1}$$

and the function $\text{lse}(\beta, z)$ is the log-sum-exponential, defined as $\text{lse}(\beta, z) := \log\left(\sum_{\mu=1}^M \exp\{\beta z_\mu\}\right)/\beta$ for any $z \in \mathbb{R}^M$ and $\beta > 0$. When applied to a sequence of input queries $X = [x_1, \ldots, x_L] \in \mathbb{R}^{d \times L}$, Eq. (1) becomes: $Z := [x_1^{\text{new}}, \ldots, x_L^{\text{new}}] = \mathcal{T}_{\text{Dense}}(X)$, and hence

$$\mathcal{T}_{\text{Dense}}(X) = \overbrace{\Xi}^{d \times M} \cdot \text{Softmax}(\beta \underbrace{\overbrace{\Xi^\top}^{M \times L} \underbrace{X}}_{M \times d \ \ d \times L}) \in \mathbb{R}^{d \times L},$$

where $\text{Softmax}(\cdot)$ performs column-wise normalization. We assume that $d = L^{o(1)}$, meaning the growth rate of $d$ is sub-polynomial relative to $L$.

Modern Hopfield Networks with continuous states are compatible with deep learning models due to their differentiable nature and the ability to retrieve patterns in a single update step. This aligns with the behavior of deep learning layers, which typically activate only once. These properties make Modern Hopfield Networks suitable as specialized layers to add memory functionalities to deep networks.

**Definition 3.6** (Hopfield attention matrix, (Ramsauer et al., 2021))**.** *The Hopfield attention matrix* $A \in \mathbb{F}_p^{n \times n}$ *is defined using model weights* $W_Q, W_K \in \mathbb{F}_p^{d \times d}$, *query patterns* $R \in \mathbb{F}_p^{n \times d}$, *and key patterns* $Y \in \mathbb{F}_p^{n \times d}$. *For* $i, j \in [n]$, *the elements of* $A$ *are given by:*

$$A_{i,j} := \exp(\beta \cdot R_{i,*} W_Q W_K^\top Y_{j,*}^\top)$$

$\mathbb{F}_p$ is the set of all $p$-bit floating-point numbers. The floating-point number set $\mathbb{F}_p$, and their operations in our computational framework are formally defined in Appendix 3.2. The Hopfield attention matrix is used to compute a single Hopfield layer.

**Definition 3.7** (Hopfield layer, page 6 in (Ramsauer et al., 2021))**.** *A single Hopfield layer propagates patterns using query patterns* $R$ *and key patterns* $Y$. *In its most general form, the result patterns* $Z \in \mathbb{F}_p^{n \times d}$ *are a function of raw stored patterns* $Y \in \mathbb{F}_p^{n \times d}$, *raw state patterns* $R \in \mathbb{F}_p^{n \times d}$, *and projection matrices* $W_Q, W_K, W_V \in \mathbb{F}_p^{d \times d}$:

$$Z = \text{softmax}(\beta \cdot R W_Q W_K^\top Y^\top) Y W_K W_V \tag{2}$$

*We set* $D := \text{diag}(\beta \cdot A \mathbf{1}_n) \in \mathbb{F}_p^{n \times n}$. *Then, based on Definition 3.6, we define the $i$-th Hopfield layer as*

$$\text{Hop}_i(R, Y_i) := D^{-1} A Y_i \widetilde{W}_V,$$

*where we denote*

$$\widetilde{W}_V = W_K W_V. \tag{3}$$

*Here, the rank of* $\widetilde{W}_V$ *is limited by dimension constraints of the matrix product* $W_K W_V$. *To provide the Hopfield layer with more flexibility, the matrix product* $W_K W_V$ *can be replaced by one parameter matrix. In this case,* $\widetilde{W}_V$ *is not the product from Eq. (3) but a stand-alone parameter matrix as in the original transformer setting.*

Multiple Hopfield layers can be integrated with other elements to build a functional Hopfield architecture.

**Definition 3.8** (Multi-layer Modern Hopfield Networks)**.** *Consider a model consisting of $m$ Hopfield layers. For each $i \in [m]$, let $f_i$ denote the components of the network excluding the $i$-th Hopfield layer, where $f_i : \mathbb{F}_p^{n \times d} \to \mathbb{F}_p^{n \times d}$. Denote the $i$-th Hopfield layer as $\text{Hop}_i$. Let the input matrix*

*or query patterns be denoted by $R \in \mathbb{F}_p^{n \times d}$, and the stored patterns (keys) associated with the $i$-th Hopfield layer by $Y_i \in \mathbb{F}_p^{n \times d}$. The multi-layer Modern Hopfield Network, denoted as $\mathsf{MHN} : \mathbb{F}_p^{n \times d} \to \mathbb{F}_p^{n \times d}$, is defined as:*

$$\mathsf{MHN}(R) = f_m \circ \mathsf{Hop}_m(f_{m-1} \circ \mathsf{Hop}_{m-1}(\cdots f_1 \circ \mathsf{Hop}_1(f_0(R), Y_1) \cdots, Y_{m-1}), Y_m),$$

*where $\circ$ denotes the composition of functions.*

We now present one type of $f_i$ function, specifically a two-layer ReLU feedforward neural network.

**Definition 3.9** (Two-layer ReLU Feed-forward Neural Networks). *We use $X \in \mathbb{F}_p^{n \times d}$ to denote the input matrix of size $n \times d$. For each $i \in [n]$, the two-layer ReLU Feed-forward Neural Networks is defined as follows:*

$$g^{\mathrm{FNN}}(X)_{i,*} := \underbrace{W_2}_{d \times d} \cdot \mathsf{ReLU}(\underbrace{W_1}_{d \times d} \cdot \underbrace{X_{i,*}}_{d \times 1} + \underbrace{b_1}_{d \times 1}) + \underbrace{b_2}_{d \times 1}.$$

### 3.4 Kernelized Hopfield Networks

The capacity of modern Hopfield networks is suboptimal. To improve the capacity, (Wu et al., 2024a) introduces a kernel as a learnable similarity measure, using stored memory patterns as training data to enhance memory capacity. Specifically, they propose the kernelized Hopfield network (KHM) defined by following update rule and energy function:

$$\mathcal{T}_\Phi(x) := \Xi \cdot \mathrm{Softmax}(\beta \mathcal{K}(\Xi, x))$$

$$E_\mathcal{K}(x) := \frac{1}{2}\mathcal{K}(x, x) + \mathrm{lse}(\beta, \mathcal{K}(\Xi, x)),$$

where the kernel $\mathcal{K}(\cdot, \cdot) := \langle \Phi(\cdot), \Phi(\cdot) \rangle : \mathbb{R}^d \times \mathbb{R}^d \to \mathbb{R}$ is associated with a learnable feature map $\Phi : \mathbb{R}^d \to \mathbb{R}^{D_\Phi}$. Here, $\mathcal{K}(\cdot, \cdot)$ acts column-wise on matrix: $\mathcal{K}(\Xi, x) = [\{\mathcal{K}(\xi_\mu, x)\}_{\mu=1}^M] = [\{\langle \Phi(\xi_\mu), \Phi(x) \rangle\}_{\mu=1}^M] \in \mathbb{R}^M$. Accordingly, we define the kernelized Hopfield layer.

**Definition 3.10** (Kernelized attention matrix). *Let $R \in \mathbb{F}_p^{n \times d}$ be the set of query (state) patterns, and $Y \in \mathbb{F}_p^{n \times d}$ be the set of key (stored) patterns. Let $W_Q \in \mathbb{F}_p^{d \times D_\Phi}, W_K \in \mathbb{F}_p^{d \times D_\Phi}$, and $W_V \in \mathbb{F}_p^{d \times d}$ be learnable projection matrices. Consider a feature map $\Phi : \mathbb{F}_p^{D_\Phi} \to \mathbb{F}_p^{D_\Phi}$ associated with a kernel function $\mathcal{K} : \mathbb{F}_p^{D_\Phi} \times \mathbb{F}_p^{D_\Phi} \to \mathbb{F}_p$ defined by $\mathcal{K}(\cdot, \cdot) := \langle \Phi(\cdot), \Phi(\cdot) \rangle$. Let $\beta > 0$ be a scaling parameter. According to Definition 3.6, the kernelized Hopfield attention matrix $A \in \mathbb{F}_p^{n \times n}$ is defined by, for every $i, j \in [n]$, $A_{i,j} := \exp(\beta \cdot \langle \Phi(R_i W_Q), \Phi(Y_j W_K) \rangle)$*

The kernelized Hopfield attention matrix is the basis for computing a single kernelized Hopfield layer.

**Definition 3.11** (Single kernelized Hopfield layer). *The result pattern $Z \in \mathbb{F}_p^{n \times d}$ are a function of raw stored patterns $Y \in \mathbb{F}_p^{n \times d}$, raw state pattern $R \in \mathbb{F}_p^{n \times d}$, and projection matrices $W_Q, W_K, W_V \in \mathbb{F}_p^{d \times d}$(For simplicity, we denote $\widetilde{W}_V$ in Hopfield layer as $W_V$):*

$$Z = \mathsf{softmax}(\beta \cdot \langle \Phi(RW_Q), \Phi(YW_K) \rangle)YW_V$$

*Note that the softmax is applied row-wise, $\langle \Phi(RW_Q), \Phi(YW_K) \rangle_{i,j} = \langle \Phi(R_i W_Q), \Phi(Y_j W_K) \rangle$. We set $D := \mathrm{diag}(\beta \cdot A1_n) \in \mathbb{F}_p^{n \times n}$. Then, based on Definition 3.6, we define the $i$-th kernelized Hopfield layer as $\mathsf{KHop}_i(R, Y_i) := D^{-1}AY_i W_V$.*

Multiple kernelized Hopfield layers can be integrated with additional components to construct the kernelized Hopfield network.

**Definition 3.12** (Kernelized Hopfield network). *We use $m$ to denote the number of kernelized Hopfield layers in the network. For $i \in \{0, 1, 2, \ldots, m\}$, we use $f_i$ to denote the other components of $i$-th kernelized Hopfield layer, where $f_i : \mathbb{F}_p^{n \times d} \to \mathbb{F}_p^{n \times d}$. Let $\mathsf{KHop}_i$ denote the $i$-th kernelized Hopfield layer. Define $R \in \mathbb{F}_p^{n \times d}$ as the state (query) patterns or input matrix, and $Y_i \in \mathbb{F}_p^{n \times d}$ as the stored (key) patterns in the $i$-th kernelized Hopfield layer. The $m$-layer kernelized Hopfield network $\mathsf{KHN} : \mathbb{F}_p^{n \times d} \to \mathbb{F}_p^{n \times d}$ is defined as:*

$$\mathsf{KHN}(R) = f_m \circ \mathsf{KHop}_m(f_{m-1} \circ \mathsf{KHop}_{m-1}(\cdots f_1 \circ \mathsf{KHop}_1(f_0(X), Y_1) \cdots, Y_{m-1}), Y_m),$$

*where $\circ$ denotes the composition of functions.*

## 4 COMPLEXITY OF MODERN HOPFIELD NETWORKS

This section explores key results regarding the circuit complexity of the computations involved in modern Hopfield networks. We begin with an analysis of the Hopfield attention matrix in Section 4.1. In Section 4.2, we delve into the computation of a single Hopfield layer. Section 4.3 extends the discussion to other components beyond the Hopfield layer. In Section 4.4, we examine the modern Hopfield network. Finally, Section 4.5 presents our main result: the circuit complexity bounds for modern Hopfield networks. These findings form the basis for our main theorem on the expressive power of Hopfield networks. In Appendix C, we provide fundamental definitions from circuit complexity theory.

### 4.1 COMPUTING HOPFIELD ATTENTION MATRIX

We first recall that matrix multiplication of two matrices is in $\mathsf{TC}^0$.

**Lemma 4.1** (Matrix multiplication in $\mathsf{TC}^0$, Lemma 4.2 in (Chen et al., 2024)). *Let $p \leq \mathrm{poly}(n)$, $n_1, n_2 \leq \mathrm{poly}(n)$, and $d \leq n$. Let $A \in \mathbb{F}_p^{n_1 \times d}$ and $B \in \mathbb{F}_p^{d \times n_2}$. Then the matrix product $AB$ can be implemented using a $\mathsf{DLOGTIME}$-uniform threshold circuit which has depth $(d_{\mathrm{std}} + d_\oplus)$ and size bounded by $\mathrm{poly}(n)$.*

Here, we extend the matrix operations to compute the Hopfield attention matrix.

**Lemma 4.2** (Computation of Hopfield attention matrix in $\mathsf{TC}^0$). *Let precision $p \leq \mathrm{poly}(n)$. One can simulate the Hopfield attention matrix $A$ using a $\mathsf{DLOGTIME}$-uniform threshold circuit with depth $4d_{\mathrm{std}} + 3d_\oplus + d_{\mathrm{exp}}$ and size bounded by $\mathrm{poly}(n)$.*

*Proof.* To compute $A_{i,j}$, we use the following steps:

1. By Lemma 4.1, we can compute the matrix product $W_Q W_K^\top$ by a $\mathsf{DLOGTIME}$-uniform circuit which has size $\mathrm{poly}(n)$ and depth $d_{\mathrm{std}} + d_\oplus$.

2. By Lemma 4.1, we can the scalar

$$s_{i,j} = R_{i,*}(W_Q W_K^\top) Y_{j,*}^\top$$

by a $\mathsf{DLOGTIME}$-uniform circuit which has size $\mathrm{poly}(n)$ and depth $2(d_{\mathrm{std}} + d_\oplus)$, following Lemma 4.1.

3. The term $\beta \cdot s_{i,j}$ is computed with depth $d_{\mathrm{std}}$ using Part 1 of Lemma 3.3.

4. Using Lemma 3.4, the exponential function $\exp(\beta \cdot s_{i,j})$ is computable with depth $d_{\mathrm{exp}}$.

Summing the circuit depths for all steps, the total depth required for $A_{i,j}$ is $d_{\mathrm{total}} = 4d_{\mathrm{std}} + 3d_\oplus + d_{\mathrm{exp}}$. Since all entries of $A$ can be simulated in parallel, the total depth $4d_{\mathrm{std}} + 3d_\oplus + d_{\mathrm{exp}}$ and size $\mathrm{poly}(n)$, completing the proof. $\qquad\square$

### 4.2 COMPUTING SINGLE HOPFIELD LAYER

This section analyzes the computation of a single Hopfield layer, including the necessary depth and size requirements.

**Lemma 4.3** (Computation of Hopfield layer in $\mathsf{TC}^0$). *Let precision $p \leq \mathrm{poly}(n)$. We can simulate the Hopfield layer using a $\mathsf{DLOGTIME}$-uniform threshold circuit with depth $8d_{\mathrm{std}} + 6d_\oplus + d_{\mathrm{exp}}$ and size bounded by $\mathrm{poly}(n)$.*

*Proof.* The computation involves multiplying matrices $D^{-1}, A, Y, \widetilde{W}_V$. First, we compute $D^{-1}$ and $A$:

**Part 1.** $D = \mathrm{diag}(\beta \cdot A\mathbf{1}_n)$ is computed with depth $d_\oplus + d_{\mathrm{std}}$, as shown in Part 1 and Part 3 of Lemma 3.3. If we compute $D^{-1}$, it requires a depth of $d_\oplus + 2d_{\mathrm{std}}$ follows from Part 1 of Lemma 3.3.

**Part 2.** From Lemma 4.2, $A$ is computed with depth $4d_{\mathrm{std}} + 3d_\oplus + d_{\mathrm{exp}}$.

**Part 3.** Then, we can multiply $A, Y$ and $\widetilde{W}_V$, which can be computed by a depth $2(d_{\text{std}} + d_\oplus)$, size $\text{poly}(n)$ uniform threshold circuit following from Lemma 4.1.

**Part 4.** Finally, the division $D^{-1} \cdot (AY\widetilde{W}_V)$ is computed in parallel with depth $d_{\text{std}}$. Combining these depths, $d_{\text{total}} = 8d_{\text{std}} + 6d_\oplus + d_{\exp}$. The total size of the circuit remains $\text{poly}(n)$, completing the proof. $\square$

### 4.3 COMPUTING COMMON COMPONENTS LAYERS

Definition 3.8 introduces modern Hopfield networks, which incorporate the Hopfield layer along with other modules such as layer normalization and two-layer ReLU feed-forward networks. In this section, we present the computational complexity of these additional components.

We begin by analyzing the circuit complexity of the two-layer ReLU feed-forward networks.

**Lemma 4.4** (Computation of Two-layer ReLU Feed-forward Networks in $\mathsf{TC}^0$). *Let $p \leq \text{poly}(n)$, one can simulate the two-layer ReLU feed-forward neural networks using a* $\mathsf{DLOGTIME}$*-uniform threshold circuit which has depth $4d_{\text{std}} + 3d_\oplus$ and size bounded by $\text{poly}(n)$.*

*Proof.* For each $i \in [n]$, Lemma 4.1 guarantees that the computation of $W_1 \cdot X_{i,*}$ requires a $\mathsf{DLOGTIME}$-uniform circuit which has depth $d_{\text{std}} + d_\oplus$ and size $\text{poly}(n)$. Applying Part 1 of Lemma 3.3, we can compute $W_1 \cdot X_{i,*} + b_1$ with an additional $\mathsf{DLOGTIME}$-uniform circuit which has depth $d_{\text{std}}$ and size $\text{poly}(n)$. The ReLU activation, $\text{ReLU}(W_1 \cdot X_{i,*} + b_1)$, also requires depth $d_{\text{std}}$ and size $\text{poly}(n)$ as per Part 1 of Lemma 3.3.

For the second layer, the circuit depth needed is $2d_{\text{std}} + d_\oplus$, and the size remains $\text{poly}(n)$. Thus, the total circuit depth across both layers is $4d_{\text{std}} + 3d_\oplus$, with the size still bounded by $\text{poly}(n)$ because these computations are able to be executed in parallel for each $i \in [n]$. $\square$

### 4.4 COMPUTING MODERN HOPFIELD NETWORKS

We demonstrate how to compute modern Hopfield networks within the $\mathsf{TC}^0$ complexity class.

**Lemma 4.5** (Computation of Modern Hopfield Networks in $\mathsf{TC}^0$). *Suppose that for every $i \in [m]$, then we can simulate function $f_i$ in* $\mathsf{MHN}$ *by a* $\mathsf{DLOGTIME}$*-uniform threshold circuit which has size $\text{poly}(n)$ and constant depth $d_f$. When $p \leq \text{poly}(n)$, the overall computation of the network can be implemented using a* $\mathsf{DLOGTIME}$*-uniform threshold circuit with depth $(m + 1)d_f + 8md_{\text{std}} + 6md_\oplus + md_{\exp}$ and size bounded by $\text{poly}(n)$.*

*Proof.* By assumption, we can simulate $f_i$ using a $\mathsf{DLOGTIME}$-uniform threshold circuit which has size bounded by $\text{poly}(n)$ and depth $d_f$.

Next, by Lemma 4.3, we know that the computation of a single Hopfield layer $\mathsf{Hop}_i$ can be performed using a $\mathsf{DLOGTIME}$-uniform threshold circuit with depth $8d_{\text{std}} + 6d_\oplus + d_{\exp}$, while its size remains polynomial in $n$.

The complete computation of $\mathsf{MHN}(R)$ involves $g_0, g_1, \ldots, g_m$ and $\mathsf{Hop}_1, \mathsf{Hop}_2, \ldots, \mathsf{Hop}_m$. The total circuit depth is, therefore, the sum of the depths of all these computations. Explicitly, the depth is $(m + 1)d_f + 8md_{\text{std}} + 6md_\oplus + md_{\exp}$. $\square$

### 4.5 MAIN RESULT: CIRCUIT COMPLEXITY BOUND OF MODERN HOPFIELD NETWORKS

Our main result establishes the circuit complexity bounds for modern Hopfield networks.

**Theorem 4.6** (Circuit complexity of modern Hopfield networks). *Consider a modern Hopfield network* $\mathsf{MHN}$ *where we can simulate each component function $f_i$ for $i \in [m]$ by a* $\mathsf{DLOGTIME}$*-uniform threshold circuit which has $d_f$ depth and $\text{poly}(n)$ size. If precision $p \leq \text{poly}(n)$, hidden dimension $d \leq O(n)$, and number of layers $m = O(1)$, then we can simulate* $\mathsf{MHN}$ *by a* $\mathsf{DLOGTIME}$*-uniform circuit family in* $\mathsf{TC}^0$.

*Proof.* Given $m = O(1)$, Lemma 4.3 establishes that the circuit depth required for $\mathsf{MHN}(R)$ is given by:

$$(m + 1)d_f + 8md_{\mathrm{std}} + 6md_\oplus + md_{\exp}.$$

Since $m$ is a constant, the total circuit depth simplifies to $O(1)$. Furthermore, the size of the circuit remains within $\mathrm{poly}(n)$.

Hence, the modern Hopfield network can be realized using a DLOGTIME-uniform circuit family belonging to $\mathsf{TC}^0$, completing the argument. □

Theorem 4.6 implies that unless $\mathsf{TC}^0 = \mathsf{NC}^1$, modern Hopfield networks with $\mathrm{poly}(n)$-precision, constant-depth, $\mathrm{poly}(n)$-size can be simulated by a DLOGTIME-uniform $\mathsf{TC}^0$ circuit family. It means that although modern Hopfield networks gain success empirically, it still suffers fundamental expressivity limitations under circuit complexity.

## 5 Complexity of Kernelized Hopfield Networks

In this section, we extend the complexity results of modern Hopfield networks to kernelized Hopfield networks. The formal definition of kernelized Hopfield networks is provided in Section 3.4.

Section 5.1 analyzes the computation of the kernelized Hopfield attention matrix. In Section 5.2, we examine the computation of a single kernelized Hopfield layer. Section 5.3 details the computation of the entire kernelized Hopfield network. Finally, in Section 5.4, we present the main results on the circuit complexity bounds for kernelized Hopfield networks. In appendix E, we provide the complete proofs for this section.

### 5.1 Computing Kernelized Hopfield Attention Matrix

In this section, we generalize the computing of the Hopfield attention matrix to the kernelized Hopfield attention matrix.

**Lemma 5.1** (Computation of kernelized Hopfield attention matrix in $\mathsf{TC}^0$). *Assuming that $p \leq \mathrm{poly}(n)$ and the linear affine feature map $\Phi$ is defined as $\Phi(u) := Wu$ for $u, v \in \mathbb{F}_p^d$, where $W \in \mathbb{F}_p^{D_\Phi \times d}$ and the linear kernel is $\mathcal{K}(u, v) = u^\top W^\top W v$, then the kernelized Hopfield attention matrix $A$ can be implemented using a DLOGTIME-uniform threshold circuit of depth $3d_{\mathrm{std}} + 2d_\oplus + d_{\exp}$ and size bounded by $\mathrm{poly}(n)$.*

### 5.2 Computing Single Kernelized Hopfield Layer

This section provides an analysis of the computation of a single kernelized Hopfield layer, including its circuit depth.

**Lemma 5.2** (Computation of Single kernelized Hopfield layer in $\mathsf{TC}^0$). *Assuming that $p \leq \mathrm{poly}(n)$, then the kernelized Hopfield layer can be implemented using a DLOGTIME-uniform threshold circuit of depth $10d_{\mathrm{std}} + 8d_\oplus + d_{\exp}$ and size bounded by $\mathrm{poly}(n)$.*

### 5.3 Computing Kernelized Hopfield Networks

Here, we analyze the computation of the entire kernelized Hopfield network.

**Lemma 5.3** (Kernelized Hopfield networks computation in $\mathsf{TC}^0$). *Assume that for each $i \in [m]$, we can simulate the function $f_i$ in KHN by a DLOGTIME-uniform threshold circuit which has constant depth $d_f$ and size $\mathrm{poly}(n)$. If the precision $p \leq \mathrm{poly}(n)$, then we can simulate the kernelized Hopfield networks by a DLOGTIME-uniform threshold circuit of depth $(m + 1)d_f + 10md_{\mathrm{std}} + 8md_\oplus + md_{\exp}$ and size bounded by $\mathrm{poly}(n)$.*

### 5.4 Main Result: Circuit Complexity Bound of Kernelized Hopfield Networks

We now present the main result for kernelized Hopfield networks, establishing their circuit complexity bounds.

**Theorem 5.4** (Main result, Circuit complexity bound of kernelized Hopfield networks). *Assume that for each $i \in [m]$, the function $f_i$ in KHN is computable by a DLOGTIME-uniform threshold circuit with constant depth $d_f$ and size $\mathrm{poly}(n)$. If precision $p \leq \mathrm{poly}(n)$, hidden dimension $d \leq O(n)$, and number of layers $m \leq O(1)$, then we can simulate the kernelized Hopfield networks KHM by a DLOGTIME-uniform circuit family within $\mathsf{TC}^0$.*

*Proof.* See Appendix E.4 for the complete proof. □

Theorem 5.4 demonstrates that, unless $\mathsf{TC}^0 = \mathsf{NC}^1$, kernelized Hopfield networks with polynomial precision, constant depth, and polynomial size can be implemented by a DLOGTIME-DLOGTIME-uniform circuit family within $\mathsf{TC}^0$. This implies that kernelized Hopfield networks are subject to inherent expressivity limitations under circuit complexity constraints despite their empirical success.

# 6 HARDNESS

In this section, we present two computational problems along with their corresponding hardness results. In In Section 6.1, we outline the corresponding hardness results. In Appendix D, we introduce the undirected graph connectivity problem and the tree isomorphism problem.

## 6.1 HARDNESS RESULTS

In this section, we present our hardness results for modern and kernelized Hopfield networks.

**Theorem 6.1.** *Assume that $\mathsf{TC}^0 = \mathsf{NC}^1$. There is no $\mathrm{poly}(n)$-precision modern Hopfield networks with constant layers, and $O(n)$ hidden dimension can solve the undirected graph connectivity problem.*

*Proof.* This result is obtained by integrating Theorem 4.6 (the circuit complexity bound of modern Hopfield networks), Lemma D.4 (showing that undirected graph connectivity is $\mathsf{NC}^1$-complete), and Fact D.3. □

**Theorem 6.2.** *Assume that $\mathsf{TC}^0 = \mathsf{NC}^1$. There is no $\mathrm{poly}(n)$-precision modern Hopfield networks with $O(1)$ layers, and $O(n)$ hidden dimension can solve the tree isomorphism problem.*

*Proof.* This result derives from Theorem 4.6 and Lemma D.8 (showing that tree isomorphism is $\mathsf{NC}^1$-complete). □

**Theorem 6.3.** *Assume that $\mathsf{TC}^0 = \mathsf{NC}^1$. There is no $\mathrm{poly}(n)$-precision kernelized Hopfield networks with constant layers, and $O(n)$ hidden dimension can solve the undirected graph connectivity problem.*

*Proof.* This result is a direct consequence of Theorem 5.4 and Lemma D.4. □

**Theorem 6.4.** *Assume that $\mathsf{TC}^0 = \mathsf{NC}^1$. There is no $\mathrm{poly}(n)$-precision kernelized Hopfield networks with constant layers, and $O(n)$ hidden dimension can solve the tree isomorphism problem.*

*Proof.* This result is a direct consequence of Theorem 5.4 and Lemma D.8. □

# 7 CONCLUSION

In this study, we conduct a comprehensive theoretical investigation of modern Hopfield networks and kernelized Hopfield networks, establishing key limitations on their computational power. Our analysis focuses on the circuit complexity of individual architectural components, showing that these networks can be simulated using uniform $\mathsf{TC}^0$ circuits. Crucially, we prove that unless $\mathsf{TC}^0 = \mathsf{NC}^1$, both modern and kernelized Hopfield networks, when implemented with polynomial precision, a constant number of layers, and hidden dimensions satisfying $d \leq O(n)$, are unable to solve problems such as undirected graph connectivity and tree isomorphism.

ETHIC STATEMENT

This paper does not involve human subjects, personally identifiable data, or sensitive applications. We do not foresee direct ethical risks. We follow the ICLR Code of Ethics and affirm that all aspects of this research comply with the principles of fairness, transparency, and integrity.

REPRODUCIBILITY STATEMENT

We ensure reproducibility of our theoretical results by including all formal assumptions, definitions, and complete proofs in the appendix. The main text states each theorem clearly and refers to the detailed proofs. No external data or software is required.

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

# Appendix

**Roadmap.** In Section A, we give an extended discussion of related works. In Section B, we present the additional theoretical results of the modern Hopfield model with a chain of thought (CoT). In Section C, we introduce essential definitions for circuit complexity used in this paper. In Section D, we present the formulation of graph connectivity and tree isomorphism problems. In Section E, we present the formal proofs for kernelized Hopfield networks.

## A    MORE RELATED WORKS

**Theory of Modern Hopfield Networks**   The Hopfield network, introduced by John Hopfield (Hopfield, 1982; 1984; Krotov & Hopfield, 2016; 2018; 2020), was originally conceived as a model for content-addressable memory. This network operates by encoding patterns through the connections between neurons, enabling associative recall. By fixing specific inputs and allowing the network to evolve dynamically, it minimizes an energy function to retrieve stored patterns. A notable characteristic of Hopfield networks is the overlapping nature of different patterns, where many patterns share common neurons. Recent advancements in Modern Hopfield Networks have showcased both theoretical significance and practical applications. These models have proven instrumental in offering insights into transformer attention mechanisms and the architecture of Transformers. For instance, (Hu et al., 2023; Wu et al., 2024a) introduced a comprehensive framework for analyzing and deriving modern Hopfield models, employing entropic regularizers to unify diverse variants. This framework encompasses sparse and generalized sparse models and includes the standard modern Hopfield network (Ramsauer et al., 2021) as a particular instance. Moreover, contemporary research efforts (Hu et al., 2023; Wu et al., 2024b; Hu et al., 2024c;a;b;d) continue to explore innovative directions for improving the efficiency and scalability of Hopfield-based models. These studies underline the transformative potential of Hopfield-driven methodologies in shaping future designs, highlighting their adaptability to complex computational tasks and their integration into broader machine learning paradigms.

**Complexity and Neural Network**   Complexity theory offers a formal framework to analyze the computational power and limitations of models. Circuit complexity, a key domain within this field, characterizes the power of Boolean circuits and has recently been applied to investigate the expressive capacity of deep neural networks and Transformers (Merrill et al., 2022; Liu et al., 2022; Merrill & Sabharwal, 2023b;a; Li et al., 2024; Chiang, 2024). Several important circuit complexity classes play a central role in machine learning. For example, $\mathsf{AC}^0$ includes problems solvable with highly parallelizable logic gates, $\mathsf{TC}^0$ extends this class by incorporating *threshold gates*, and $\mathsf{NC}^1$ describes languages recognized by circuits with $O(\log n)$-depth and bounded fan-in (Merrill et al., 2022). These classes exhibit the hierarchy $\mathsf{AC}^0 \subset \mathsf{TC}^0 \subseteq \mathsf{NC}^1$, although whether $\mathsf{TC}^0 \neq \mathsf{NC}^1$ remains an open question.

Transformers' depth requirements have been analyzed through the lens of these complexity classes. For instance, (Liu et al., 2022) demonstrates that the depth of a Transformer must scale with input length to simulate certain non-solvable semi-automata. In a similar vein, (Li et al., 2024) investigates the connection between constant-depth Transformers, chain-of-thought (CoT) reasoning, and circuit complexity. They show that $\mathsf{T}[\mathrm{poly}(n), 1, 1] \subseteq \mathsf{CoT}[\log n, \mathrm{poly}(n), 1, 1] \subseteq \mathsf{AC}^0$, and $\mathsf{T}[\mathrm{poly}(n), \log n, 0] \subseteq \mathsf{CoT}[\log n, \mathrm{poly}(n), \log n, 0] \subseteq \mathsf{TC}^0$, Here, $\mathsf{T}[d(n), s(n), e(n)]$ denotes the constant-depth Transformers characterized by precision of $s(n)$, an embedding size of $d(n)$, and $e(n)$ exponent bits, while $\mathsf{CoT}[T(n), d(n), s(n), e(n)]$ refers to a chain-of-thought (CoT) Transformer operating for $T(n)$ steps. These findings highlight that intermediate reasoning steps in CoT models can improve computational expressivity.

The Strong Exponential Time Hypothesis (SETH), which posits that no $k$-SAT algorithm exists with runtime $O(2^{(1-\epsilon)n})$ for $\epsilon > 0$ and $k \geq 3$, serves as a cornerstone in fine-grained complexity analyses (Impagliazzo & Paturi, 2001). Results derived from SETH have shaped Transformer-related studies, particularly for tensor attention mechanisms (Alman & Song, 2023a; 2024; Liang et al., 2024b;a; Alman & Song, 2023b). For example, (Alman & Song, 2023a) shows that by assuming SETH, it is impossible to perform forward pass computation of attention-based networks in $O(n^2)$

time, while (Alman & Song, 2024) extends this limitation to backward computations. These results underscore the intrinsic computational challenges of Transformer architectures.

**Limitations of Transformer Architectures.** While Transformers excel in natural language processing, their capabilities in mathematical computation remain constrained (Charton, 2022). To understand these limitations, researchers have examined two types of Transformers: (1) average-head attention, which assigns 1 to the largest probability entry and 0 to others, and (2) softmax-attention, which computes probabilities as $\mathsf{Softmax}(X) = \mathrm{diag}(\exp(X) \cdot \mathbf{1}_n)^{-1} \cdot \exp(X)$. Average-head Transformers can recognize languages beyond $\mathsf{AC}^0$, but they are confined to $\mathsf{TC}^0$, as shown by (Merrill et al., 2022). Similarly, softmax-attention Transformers are also within $\mathsf{TC}^0$ (Liu et al., 2022). Extending this, (Merrill & Sabharwal, 2023a) formalize a generalized similarity function $s$ and demonstrate that softmax-attention Transformers fall under L-uniform $\mathsf{TC}^0$. Through a first-order logic framework with MAJORITY quantifiers (FOM) (Immerman, 1998), (Merrill & Sabharwal, 2023b) show that these Transformers can be simulated in DLOGTIME-uniform $\mathsf{TC}^0$. (Chiang, 2024) further enhances precision, proving that Transformers with minimal error ($2^{-O(\mathrm{poly}(n))}$) also belong to this class. For practical problems, (Feng et al., 2023) reveal that unless $\mathsf{TC}^0 = \mathsf{NC}^1$, log-precision Transformers cannot solve arithmetic, equation-solving, or Context-Free Grammar (CFG) membership tasks (Sipser, 1996). These findings explain why Transformers often struggle with mathematical reasoning. Recently, (Chen et al., 2024) studied the circuit complexity bounds of RoPE-based transformer architectures.

# B    Additional Theoretical Results for Modern Hopfield Model with Chain of Thought

We provide theoretical results for modern Hopfield networks that incorporate the chain of thought (CoT) mechanism. We first describe the architecture of the modern Hopfield model.

## B.1    Modern Hopfield Model Architecture with CoT

Let $\mathcal{V}$ be the vocabulary, and consider a Hopfield model with parameters $\theta$ and a maximum input length $n_{\max}$. This model maps an input token sequence $(x_1, \ldots, x_n) \in \mathcal{V}^n$ (for $n \leq n_{\max}$) to a probability distribution over $\mathcal{V}$, and we denote it by $p_\Theta(\cdot | x_1, \ldots, x_n)$. We use $\mathsf{MHM}_\theta(x)$ as the function that selects the token in $\mathcal{V}$ which can maximize $p_\Theta(\cdot | x_1, \ldots, x_n)$, defined as: $\mathsf{MHM}_\theta(x_1, \ldots, x_n) := \arg\max_{y \in \mathcal{V}} p_\theta(y | x_1, \ldots, x_n)$.

We define the Next-token generator with parameters $\theta$ and a maximum input length $n_{\max}$ maps sequences from $\cup_{n=1}^{n_{\max}} \mathcal{V}^n$ to $\mathcal{V}$. It is recursively defined as: $\mathsf{MHM}_\theta^i(x_1, x_2, \ldots, x_n) = \mathsf{MHM}_\theta^{i-1}(x_1, x_2, \ldots, x_n, \mathsf{MHM}_\theta^{i-1}(x_1, x_2, \ldots, x_n))$.

For every $i \geq 1$, provided $i + n \leq n_{\max} - 1$, we have the base case: $\mathsf{MHM}_\theta^1(x_1, x_2 \ldots, x_n) = \mathsf{MHM}_\theta(x_1, x_2, \ldots, x_n)$.

Architecture Overview: The architecture is similar to GPT-like models, comprising four main components: 1. A layer used for token embedding, 2. A layer used for position encoding, 3. An output linear layer, and 4. $L$ identical decoder layers. Each decoder layer consists of a single Hopfield layer (see Definition 3.7) and a two-layer ReLU feedforward network (see Definition 3.9). Together, these decoder layers form an $L$-layer modern Hopfield network (see Definition 3.8). The model parameters $\theta$ are organized as: $\theta = (\theta_{\mathsf{PE}}, \theta_{\mathsf{TE}}, \theta_{\mathsf{OUTPUT}}, \theta_{\mathsf{MHN}})$, where $\theta_{\mathsf{MHN}} = \{\theta_{\mathsf{Hop}}^{(l)}, \theta_{\mathsf{FNN}}^{(l)}\}_{l=0}^{L-1}$ are trainable parameters (see Algorithm 1).

The token embedding layer parameterized by $\theta_{\mathsf{TE}} \in \mathbb{R}^{d \times |\mathcal{V}|}$ maps tokens in $\mathcal{V}$ to $\mathbb{R}^d$: $\theta_{\mathsf{TE}}(x), \quad \forall x \in \mathcal{V}$. The position encoding layer with parameters $\theta_{\mathsf{PE}} \in \mathbb{R}^{d \times |\mathcal{V}|}$ maps positions in $[n_{\max}]$ to $\mathbb{R}^d$: $\theta_{\mathsf{PE}}(n), \quad \forall n \in [n_{\max}]$. The output layer parameterized by $\theta_{\mathsf{OUTPUT}} \in \mathbb{R}^{|\mathcal{V}| \times d}$ is defined by: $\mathsf{OUTPUT}_{\theta_{\mathsf{OUTPUT}}}(h) = \mathrm{softmax}(\theta_{\mathsf{OUTPUT}} h), \quad \forall h \in \mathbb{R}^d$.

---

**Algorithm 1** Modern Hopfield Model, $\text{MHM}_\theta$ and $p_\theta$

---

1: Input: Parameters $\theta_{\text{PE}}, \theta_{\text{TE}}, \theta_{\text{OUTPUT}}, \theta_{\text{MHN}}$ and input tokens $(x_1, \ldots, x_n) \in \mathcal{V}^n$
2: Output: An output token $\text{MHM}_\theta(x)$, an output distribution $p_\theta(\cdot|x_1, \ldots, x_i)$ for every $i \in [n]$.
3: $h_i^{(0)} \leftarrow \theta_{\text{TE}}(x_i) + \theta_{\text{PE}}(i), \quad \forall i \in [n]$
4: $(h_1^{(1)}, \ldots, h_n^{(1)}) \leftarrow \text{MHN}_{\theta_{\text{MHN}}}(h_1^{(0)}, \ldots, h_n^{(0)})$
5: $p_\theta(\cdot|x_1, \ldots, x_i) \leftarrow \text{OUTPUT}_{\theta_{\text{OUTPUT}}}(h_i^{(1)})$ for $i \in [n]$
6: $\text{MHM}_\theta(x) \leftarrow \arg\max_y p_\theta(y|x_1, \ldots, x_n)$.

---

## B.2 RESULTS OF HOPFIELD WITH CoT

In this subsection, we present the theoretical results for the modern Hopfield model. First, we show that the Hopfield update rule can achieve an attention mechanism.

**Lemma B.1** (The Hopfield update rule can be interpreted as a form of the transformer attention mechanism, page 5 in (Ramsauer et al., 2021)). *Let $N$ be the number of stored (key) patterns, represented as row vectors $y_i$, and $S$ be the number of state (query) patterns, represented as row vectors $r_i$. Define $Y = (y_1, \ldots, y_N)^\top, R = (r_1, \ldots, r_S)^\top$. The Hopfield update rule can be expressed as:*

$$Z = \text{softmax}(\frac{1}{\sqrt{d_k}} Q K^\top) V$$
$$= \text{softmax}(\beta \, R W_Q W_K^\top Y^\top) Y W_K W_V, \tag{4}$$

*where the following hold: $X^\top = K = Y W_K, \Xi^\top = Q = R W_Q, V = Y W_K W_V = X^\top W_V$, and $W_K \in \mathbb{R}^{d_y \times d_k}, W_Q \in \mathbb{R}^{d_r \times d_k}, W_V \in \mathbb{R}^{d_k \times d_v}$, and $\beta = \frac{1}{\sqrt{d_k}}$. Here, $d_k$ is the dimension of the Hopfield space, $d_y$ and $d_r$ are the dimensions of the stored and state patterns, and $Z$ represents the attention scores.*

*The middle part of Eq. 4 corresponds to the attention mechanism of the transformer. Specifically, when $R = Y$ and $W_K W_V$ are substituted with $W_V$, the Hopfield update rule is reduced to transformer self-attention.*

**Definition B.2** (Word Problem for Group $G$, (Li et al., 2024)). *For a group $G$ and $n$ elements $(f_1, \ldots, f_n)$ from $G$, the word problem $\mathcal{L}_G$ is defined as the problem of determining whether the product $f_1 \circ f_2 \circ \cdots \circ f_n$ equals the identity element of $G$.*

**Lemma B.3** (Theorem 3.5 in (Li et al., 2024)). *Assuming $\text{TC}^0 \subsetneq \text{NC}^1$, Transformer with $n$ step CoT, $\log n$ embedding size, and constant precision can solve the wording problem $\mathcal{L}_{S_5}$, $S_5$ is permutation groups over 5 elements. However, a Transformer with constant depth, polynomial embedding size, and $\log n$ precision, but without CoT, cannot solve it.*

The proof of Theorem B.4 is a direct consequence of Lemma B.3.

**Theorem B.4.** *Assuming $\text{TC}^0 \subsetneq \text{NC}^1$, modern Hopfield model with $n$ step CoT, embedding size $\log n$, and constant precision can solve the word problem $\mathcal{L}_{S_5}$. However, the modern Hopfield model with a constant depth, polynomial embedding size $\text{poly}(n)$, and $\log n$ precision, but without CoT, cannot solve it.*

## C BACKGROUND ON CIRCUIT COMPLEXITY

**Definition C.1** (Boolean circuit). *We define a Boolean circuit that has $n$ inputs and $m$ outputs as a function $C_n : \{0, 1\}^n \rightarrow \{0, 1\}^m$ on a directed acyclic graph. Each node in the graph represents a logic gate in $\{\text{AND}, \text{OR}, \text{NOT}\}$. The graph contains $n$ input nodes with in-degree 0 and $m$ output nodes with out-degree 0. The circuit evaluates the value at each non-input gate based on the inputs it receives from other gates. The size of the circuit is the total number of gates, and its depth is the maximum length of a directed path.*

To address varying input sizes, the concept of circuit families is introduced.

**Definition C.2** (Circuit family and language recognition, Definition 1.10 on page 10 in (Vollmer, 1999)). *A circuit family is a set $\mathcal{C} = \{C_n : n \in \mathbb{N}\}$, where for each $n \in \mathbb{N}$, $C_n$ is a circuit*

with $n$ inputs. The family $\mathcal{C}$ computes a function $f : \{0, 1\}^* \rightarrow \{0, 1\}^*$ if for every $x \in \{0, 1\}^*$, $C_{|x|}(x) = f(x)$. Similarly, $\mathcal{C}$ is said to compute a language $L \subseteq \{0, 1\}^*$ if $\mathcal{C}$ computes the indicator function $\mathbb{1}_L$ of $L$.

A language $L \subseteq \{0, 1\}^*$ is a set of binary strings that represent the "yes" instances of a given decision problem, where $\{0, 1\}^*$ denotes the set of all finite binary strings of any length.

The complexity classes $\mathsf{NC}^i$, $\mathsf{AC}^i$, and $\mathsf{TC}^i$ characterize languages recognizable by Boolean circuits under different constraints. $\mathsf{NC}^i$ consists of languages that can be computed using circuits with polynomial size, depth $O((\log n)^i)$, and bounded fan-in AND, OR, and NOT gates. Extending this, $\mathsf{AC}^i$ allows unbounded fan-in for AND and OR gates while maintaining the same size and depth constraints, enabling the recognition of a broader range of languages. Further generalizing, $\mathsf{TC}^i$ incorporates MAJORITY gates. These $\mathsf{TC}^i$ circuits also support unbounded fan-in AND, OR, and NOT gates while adhering to the same size and depth limits, making them more expressive than both $\mathsf{NC}^i$ and $\mathsf{AC}^i$.

The class P encompasses languages decidable by deterministic Turing machines in polynomial time. Circuit hierarchies relate $\mathsf{NC}^i$, $\mathsf{AC}^i$, and $\mathsf{TC}^i$, showing that for any fixed $i \in [n]$,

$$\mathsf{NC}^i \subseteq \mathsf{AC}^i \subseteq \mathsf{TC}^i \subseteq \mathsf{NC}^{i+1} \subseteq \mathsf{P}.$$

However, whether $\mathsf{TC}^0$ is strictly contained in $\mathsf{NC}^1$ remains unresolved.

Uniform circuit families are more practical and relevant in computational theory compared to non-uniform families. For L-uniformity, a Turing machine with logarithmic space must output a circuit for any given input size. Alternatively, DLOGTIME-uniformity requires a random access Turing machine to construct circuits in logarithmic time. These definitions ensure feasibility and consistency in circuit design. While DLOGTIME-uniformity is generally equivalent to L-uniformity, differences arise in smaller circuit classes that lack the ability to simulate their own generation process.

# D GRAPH CONNECTIVITY AND TREE ISOMORPHISM PROBLEMS

## D.1 UNDIRECTED GRAPH CONNECTIVITY PROBLEM

**Definition D.1** (Undirected graph connectivity problem). *The undirected graph connectivity problem determines whether a path connects two specific vertices $u$ and $v$ in an undirected graph $G$.*

**Definition D.2** (FL, page 8 on (Cook, 1985)). *The class FL is the set of languages recognized by a deterministic Turing Machine in space $O(\log n)$.*

Following the definitions, we have some results.

**Fact D.3** (Proposition 4.1 in (Cook, 1985)). $\mathsf{NC}^1 \subseteq \mathsf{FL}$

**Lemma D.4** (Theorem 3 in (Cook & McKenzie, 1987)). *Undirected graph connectivity is $\mathsf{NC}^1$-hard for FL in definition D.2. When the given graph is known to be a disjoint union of cycles, the connectivity problem is $\mathsf{NC}^1$-complete for FL.*

## D.2 TREE ISOMORPHISM PROBLEM

**Definition D.5** (Colored trees, page 2 on (Jenner et al., 1998)). *A tree with n nodes is said to be colored if each node in the tree is labeled with a positive integer no greater than n.*

**Definition D.6** (Isomorphism, page 2 on (Jenner et al., 1998)). *An isomorphism between two colored trees $T_1$ and $T_2$ is a bijection between their node sets that satisfies the following conditions:*

- *The root of $T_1$ maps to the root of $T_2$.*

- *The colors of the nodes are preserved.*

- *The structure of the edges is maintained.*

*We denote $T_1 \simeq T_2$ if an isomorphism exists between them. These notions are similarly applicable to non-colored trees.*

Following the definition, we explore its computational implications.

**Definition D.7** (Tree isomorphism problem, page 2 on (Jenner et al., 1998)). *Given two trees $T_1$ and $T_2$, the tree isomorphism problem is to determine whether $T_1 \simeq T_2$. If the trees are colored, the problem is referred to as the colored tree isomorphism problem.*

**Lemma D.8** (Theorem 3.3 in (Jenner et al., 1998)). *In the string representation, the colored tree isomorphism problem and the tree isomorphism problem are $\mathsf{NC}^1$-complete under $\leq^{\mathrm{DLT}}$, where $\leq^{\mathrm{DLT}}$ denotes DLOGTIME reducibility.*

# E   PROOFS OF SECTION 5

## E.1   PROOFS OF LEMMA 5.1

**Lemma E.1** (Lemma 5.1 Restated). *Assuming that $p \leq \mathrm{poly}(n)$ and the linear affine feature map $\Phi$ is defined as $\Phi(u) := Wu$ for $u, v \in \mathbb{F}_p^d$, where $W \in \mathbb{F}_p^{D_\Phi \times d}$ and the linear kernel is $\mathcal{K}(u, v) = u^\top W^\top W v$, then the kernelized Hopfield attention matrix $A$ can be simulated using a DLOGTIME-uniform threshold circuit with depth $3d_{\mathrm{std}} + 2d_\oplus + d_{\exp}$ and size bounded by $\mathrm{poly}(n)$.*

*Proof.* We compute the entry of the kernelized attention matrix $A$:

$$A_{i,j} := \exp(\beta \cdot (WR_iW_Q)^\top (WY_jW_K)).$$

Using Lemma 4.1, the scalar product

$$s_{i,j} = (WR_iW_Q)^\top (WY_jW_K)$$

can be simulated by a DLOGTIME-uniform threshold circuit which has $\mathrm{poly}(n)$ size and $5(d_{\mathrm{std}} + d_\oplus)$ depth.

Next, by Part 1 of Lemma 3.3, the product $\beta \cdot s_{i,j}$ requires depth $d_{\mathrm{std}}$.

Finally, by Lemma 3.4, the exponential function $\exp(\beta \cdot s_{i,j})$ can be computed with depth $d_{\exp}$ and size $\mathrm{poly}(n)$.

Summing the depths, the overall depth for $A_{i,j}$ is:

$$d_{\mathrm{total}} = 6d_{\mathrm{std}} + 5d_\oplus + d_{\exp}.$$

As all $A_{i,j}$ can be computed in parallel, the attention matrix $A$ requires a DLOGTIME-uniform threshold circuit which has depth $6d_{\mathrm{std}} + 5d_\oplus + d_{\exp}$ and size $\mathrm{poly}(n)$.

$\square$

## E.2   PROOFS OF LEMMA 5.2

**Lemma E.2** (Lemma 5.2 Restated). *Assuming that $p \leq \mathrm{poly}(n)$. Then we can simulate kernelized Hopfield layer using a DLOGTIME-uniform threshold circuit with depth $10d_{\mathrm{std}} + 8d_\oplus + d_{\exp}$ and size bounded by $\mathrm{poly}(n)$.*

*Proof.* To compute a single layer, we must multiply the matrices $D^{-1}$, $A$, $Y$, and $W_V$. First, we compute $D$ and $A$. Using $D := \mathrm{diag}(\beta \cdot A1_n)$, $D$ can be simulated with depth $d_\oplus + d_{\mathrm{std}}$ and size $\mathrm{poly}(n)$ by Part 1 and Part 3 of Lemma 3.3.

From Lemma 5.1, computing $A$ requires depth $6d_{\mathrm{std}} + 5d_\oplus + d_{\exp}$.

Next, matrix products $A$, $Y$, and $W_V$ can be simulated with depth $2(d_{\mathrm{std}} + d_\oplus)$ and $\mathrm{poly}(n)$ size, as shown in Lemma 4.1. Finally, the computation of $D^{-1} \cdot (AYW_V)$ can be performed using parallel division, requiring depth $d_{\mathrm{std}}$ and size $\mathrm{poly}(n)$, which follows from Part 1 of Lemma 3.3.

Combining these results, the total depth is:

$$d_{\mathrm{total}} = 10d_{\mathrm{std}} + 8d_\oplus + d_{\exp}.$$

Since the operations can run in parallel with size $\mathrm{poly}(n)$, the kernelized Hopfield layer is simulated by DLOGTIME-uniform threshold circuit with depth $10d_{\mathrm{std}} + 8d_\oplus + d_{\exp}$. $\square$

### E.3 PROOFS OF LEMMA 5.3

**Lemma E.3** (Lemma 5.3 Restated). *Assume that, for each $i \in [m]$, we can simulate the function $f_i$ in KHN by a DLOGTIME-uniform threshold circuit with $\mathrm{poly}(n)$ size and constant depth $d_f$. Let the precision $p \leq \mathrm{poly}(n)$. Then we can simulate kernelized Hopfield networks using a DLOGTIME-uniform threshold circuit with depth $(m+1)d_f + 10md_{\mathrm{std}} + 8md_\oplus + md_{\exp}$ and size bounded by $\mathrm{poly}(n)$.*

*Proof.* By assumption, for each $i \in [m]$, we are able to simulate $f_i$ by a DLOGTIME-uniform circuit of $d_g$ depth and $\mathrm{poly}(n)$ size. From Lemma 4.3, each $\mathsf{KHop}_i$ has depth $10d_{\mathrm{std}} + 8d_\oplus + d_{\exp}$.

To compute $\mathsf{KHN}(R)$, we need circuits for $f_0, f_1, \ldots, f_m$ and $\mathsf{KHop}_1, \ldots, \mathsf{KHop}_m$. The total circuit depth becomes:

$$(m+1)d_g + 10md_{\mathrm{std}} + 8md_\oplus + md_{\exp}.$$

The circuit size remains $\mathrm{poly}(n)$, proving that the kernelized Hopfield network is computable under these conditions. $\qquad\square$

### E.4 PROOFS OF THEOREM 5.4

**Theorem E.4** (Theorem 5.4 Restated). *Assume that, for each $i \in [m]$, we can simulate the function $f_i$ in KHN by a DLOGTIME-uniform threshold circuit with constant depth $d_f$ and size $\mathrm{poly}(n)$. If precision $p \leq \mathrm{poly}(n)$, hidden dimension $d \leq O(n)$, and number of layers $m \leq O(1)$, then we can simulate kernelized Hopfield networks KHM by a DLOGTIME-uniform circuit family within $\mathsf{TC}^0$.*

*Proof.* With $m = O(1)$, Lemma 5.3 ensures that the circuit depth for $\mathsf{KHM}(R)$ is:

$$(m+1)d_g + 10md_{\mathrm{std}} + 8md_\oplus + md_{\exp} = O(1),$$

with size $\mathrm{poly}(n)$. Thus, the kernelized Hopfield network can be implemented using a uniform circuit family within $\mathsf{TC}^0$. $\qquad\square$

## LLM USAGE DISCLOSURE

LLMs were used only to polish language, such as grammar and wording. These models did not contribute to idea creation or writing, and the authors take full responsibility for this paper's content.

