# OpenReview forum: "Modern Hopfield Networks Cannot Solve $\mathsf{NC}^1$-Hard Problems"
_ICLR.cc/2026/Conference — Submitted to ICLR 2026_

### Official Review · Reviewer_cE2B · 2025-10-21

**Soundness:** 3
**Presentation:** 1
**Contribution:** 2
**Rating:** 2
**Confidence:** 4

**Summary:**

The submission studies the circuit complexity of two varieties of Hopfield Networks, specifically MHNs and KHNs. It shows that under certain basic restrictions, the expressivity of these networks in terms of circuit complexity lies between NC1 and DLOGTIME-uniform TC0.

**Strengths:**

The study of complexity-theoretic aspects of machine learning is a worthwhile endeavour, and Hopfield Networks are a well-established notion. Essentially, the research question(s) underlying the submission are well-founded and interesting.

**Weaknesses:**

As a purely theoretical submission, I find it very surprising that there is no discussion at all concerning the technical contributions - that is, the proof techniques and challenges that had to be overcome in order to achieve the claimed results. What makes these results notable and of interest from a foundational perspective? There is no discussion whatsoever about the ideas employed to obtain the results. Based on the proofs in the paper and in the appendix, it seemed to me that all the individual proofs follow standard arguments and are neither challenging nor yield truly novel or surprising insights. To me, this forms the main weakness of the submission: without a careful and understandable discussion of the contributions, I find it very difficult to recommend acceptance.

Minor weaknesses / suggestions for improvement:

-The statements in the second paragraph ("Understanding modern Hopfield networks’ computational capabilities and limitations is critical for their effective application. ...") are too strong, especially since they are not substantiated by any reference. I agree that exploring their expressiveness is a worthwhile and interesting research direction, but the tone of the 2nd paragraph should be toned down.

-A broader introduction is necessary before one can list the main contributions on page 2. Currently, there is a leap between the high-level text earlier and the sudden appearance of unexplained terms such as "precision", "hidden dimension", "depth", "layers" and "kernelized HNs" (which are different from MHNs but are not mentioned at all until suddenly appearing in the main contributions). Given the breadth of ICLR, one cannot assume that readers are familiar with Hopfield networks - the natural way to address this would have been to include one paragraph with a high-level, quick overview of what a (M)HN actually is. This high-level overview would then also make it easier for readers to digest the technical introduction of HNs in Subsections 3.3 and 3.4.

As a related point to the above, the pointer to Appendix A on page 2 should also mention that this appendix also discusses other works on circuit complexity and expressivity, and provides a brief introduction to AC0 and NC1. I feel that this part of the related work is in fact important and should appear in Section 2 (instead of the general related work on HNs or some of the proofs later on).

-The phrasing of the sentences summarizing the two main contributions on page 2 is a bit off and should be revised. For instance, the first sentence has the structure "... any ... networks ... is...", which mixes plural and singular. In the second sentence, there's an additional issue regarding the quantification of "a ... networks" - it should be "any".

**Questions:**

in spite of the many references for the claim that MHNs can replace conventional layers, the references are almost all non-peer-reviewed sources (e.g., arxiv papers). Are there no peer-reviewed references for these claims? Moreover, if MHNs were introduced only in 2021, how come the references claiming that MHNs can replace conventional layers all appeared significantly earlier than in 2021? And do similar claims also carry over to the KHNs studied in the submission?

Naturally, the authors may also comment on the listed weaknesses if they wish to do so.

---

### Official Review · Reviewer_FZQq · 2025-10-29

**Soundness:** 2
**Presentation:** 1
**Contribution:** 2
**Rating:** 2
**Confidence:** 3

**Summary:**

In this article, the authors present new results about the computational complexity of Modern Hopfield Networks.

The main contributions are:

- a positive result, showing that those Modern Hopfield Networks (including their Kernelized variant) are in a special family of circuits, called the DLOGTIME-uniform TC$^{0}$. This class corresponds to a class of functions computable by a family of threshold circuits of constant depth, polynomial size, and uniform structure (deterministic Turing machine that, given the input size $n$, and the index of a gate, can describe the structure of $C_n$ in logarithmic time).

- a negative result, showing that the same families of Modern Hopfield Networks cannot solve certain classes of problems, called NC$^{1}$-hard  (with some examples of problems not solvable, for example tree isomorphism).

**Strengths:**

- The results are interesting for the communities at the intersecion of computational complexity and machine learning.

- Modern Hopfield Networks are theoretically elegant and useful for specialized (though niche) research, progress on understanding their properties can be interesting for various communities.

**Weaknesses:**

- The second contribution (negative result) is presented as a separate contribution from the first one, but is a consequence of the first result and existing known result of the literature.  This limits in a way the substance of the contributions of the article.

- The statement of Theorems 6.1 to 6.4 start with ``Assume that TC$^{0}$=NC$^{1}$ [...]''. This seems to be in contradiction with the presentation of the main contributions stating that unless this equality holds, Modern Hopfield Networks cannot solve NC$^{1}$-hard problems.

- (minor) The authors do motivate their study in their introduction and a paragraph page 2 of Hopfield Networks, but I feel they fail to convey the (relative) importance of their usage.  (lot of references, but no comparison of perspective).

- Given the second point above, and the absence of illustrative experiments, the contributions feel limited.

**Questions:**

- Can the authors confirm or the statements of Theorems 6.1 to 6.4 with respect to the third point above?

---

### Official Review · Reviewer_iK9k · 2025-10-30

**Soundness:** 2
**Presentation:** 2
**Contribution:** 2
**Rating:** 2
**Confidence:** 3

**Summary:**

This paper provides a circuit complexity theory based theoretical analysis of Modern Hopfield Networks. Poly(n)-precision MHNs with constant depth and O(n) hidden dimension can be simulated by DLOGTIME-uniform TC-0 circuits. Similar bounds are also provided for Kernelized Hopfield Networks. It is shown that MHNs and KHNs cannot solve NC1 problems such as undirected graph connectivity and tree isomorphism unless TC0=NC1.

**Strengths:**

- The paper provides circuit complexity theoretic analysis of Modern Hopfield Networks and Kernel Hopfield Networks.
- It is shown that MHNs and KHNs cannot solve NC1 problems such as undirected graph connectivity unless TC0=NC1.

**Weaknesses:**

- Weak implications of the results: the paper presents the results without much implications. Showing that MHNs/KHNs can be simulated by DLOGTIME-uniform TC-0 circuits lacks proper implications other than stating that it cannot solve NC1 problems. The author needs to explore in depth what other theoretical or practical implications does this result have. On practical side some experiments/comparisons needs to be done with real life systems, e.g., poly(n) precision: Modern GPUs use fixed precision. How does the theory change for different assumptions that may change in practical implementations.
- Lacks novelty: the main results are highly dependent on the results from Chiang (2024) and Chen et al. (2024), Most technical lemmas (3.3, 3.4, 3.5, 4.1) are directly taken from existing works and is directly used in the MHN construct.
- Writing quality: the writing was a bit confusing to me, some basic definitions of circuit complexity needs to be provided in the main paper instead of appendix since a large group of readers in an ML conference may not be familiar with circuit complexity  theory.

**Questions:**

Please check the weaknesses if they can be addressed

---

### Official Review · Reviewer_a77z · 2025-10-31

**Soundness:** 3
**Presentation:** 2
**Contribution:** 1
**Rating:** 4
**Confidence:** 4

**Summary:**

This paper shows that Modern Hopfield Networks (MHNs) and kernel variants can be simulated by DLOGTIME-uniform $TC^0$ circuits, i.e., constant-depth, polynomial-size (in the input) threshold computation circuits. This result therefore highlights the limitations of MHNs in regard to their approximation / expressivity properties.

**Strengths:**

On the whole I think the paper is quite well written. MHNs are an interesting area and understanding them theoretically seems like a valuable line of work.

**Weaknesses:**

In a nutshell I think the claims of the paper are mostly well founded but it seems to me that there is little novelty or new insight here over the prior work (unless I am quite mistaken!). With bounded depth it just doesn't seem that surprising that an MHN, or indeed any bounded depth network, would be $TC^0$?

More in-depth version: the main drawback of the paper I think is its novelty / contribution. The paper by Chiang seems to be the source of results stating that certain primitives of networks are $TC^0$. With these established it seems straight forward that any finite composition of these primitives would be $TC^0$. As a result, the key results mostly feel like corollaries of prior work. Moreover, if this is the basic idea then why the focus just on MHNs? The proof technique doesn't seem to use anything that specific really concerning MHNs, therefore I think this focus may be obscuring the point. Once network primitives (e.g., matrix multiplication, ReLU, Softmax) are established as $TC^0$, then (roughly speaking)   would expect any bounded depth network should also be $TC^0$. What am I missing? In short, the bounded depth does a lot of the heavy lifting it seems to me here.

**Questions:**

- What would you highlight as the key technical innovations you bring over the Chiang paper?

- Why not just state your results for any model which consists of a constant depth network of $TC^0$ primitives?

- Clarity around DLOGTIME and where it applies, i.e., training versus inference. DLOGTIME-uniformity means that there exists a single deterministic algorithm which, given the input size n, can construct or describe the wiring and gate types of the circuit in O(log n) time. If a class of networks is said to be DLOGTIME-uniform, this implies that for every network size n, there is a uniform, efficiently computable procedure that produces the threshold circuit implementing it. The parameters of the network though are typically the outcome of a data-dependent optimization or training process. Therefore, before seeing any particular trained MHN, we would have to possess a single log-time algorithm capable of generating the corresponding threshold circuit, including all its weight constants, for any possible trained network. This confuses me a bit, the learned parameters could depend arbitrarily on data for instance, how are they produced by a fixed DLOGTIME procedure? Is it then that while evaluating a fixed MHN might be DLOGTIME-uniform, simulating training / a trained network from data is not?

- Do you need to also assume that your input data is bounded?

- Are your results just for linear kernels?

**Details Of Ethics Concerns:**

Not applicable.

---

### Meta-Review · Area_Chair_acJo · 2026-01-01

**Summary:**

After careful consideration of the reviewers’ evaluations, the consensus decision is to reject this submission. The primary concerns center on limited novelty, weak theoretical contribution, and insufficient contextualization of results, which collectively undermine the paper’s suitability for ICLR 2026.

The key points raised by the reviewers are:

Lack of Novelty and Technical Depth
Multiple reviewers noted that the core technical results heavily rely on prior work (e.g., Chiang 2024, Chen et al. 2024), with several lemmas directly adopted without significant new insight. The main claim—that constant-depth Modern Hopfield Networks (MHNs) are in DLOGTIME-uniform \(\text{TC}^0\)
 —was seen as a straightforward consequence of known results about network primitives being in \(\text{TC}^0\). Reviewers expressed that the paper does not clearly articulate what new technical challenges were overcome or what conceptual advances are offered beyond existing literature.

Weak Implications and Practical Relevance
Reviewers found the implications of the results underdeveloped. While the paper correctly concludes that MHNs cannot solve \(\text{NC}^1\)-hard problems under certain assumptions, it does not sufficiently explore what this means for the design, application, or limitations of MHNs in practice. There is no discussion of how assumptions like polynomial precision relate to real-world implementations (e.g., fixed-precision hardware), nor are there empirical comparisons or illustrative examples to ground the theoretical claims.

Presentation and Clarity Issues
The writing was described as confusing in places, with important concepts from circuit complexity relegated to the appendix rather than explained in the main text. Several reviewers pointed out structural issues: the contributions are presented without adequate high-level motivation; key terms (e.g., kernelized Hopfield Networks) are introduced abruptly; and the relationship between the positive and negative results is not clearly framed. One reviewer also noted grammatical and phrasing issues that affect readability.

Scope and Focus
Reviewers questioned why the analysis is restricted to MHNs when the proof techniques appear generic and applicable to any constant-depth network composed of \(\text{TC}^0\)-expressible primitives. This narrow focus limits the broader significance of the work and suggests that the results may be more about the circuit complexity of bounded-depth architectures in general, rather than offering specific insights into Hopfield networks.

Overall Assessment
While the topic is relevant and the technical statements appear sound, the paper does not meet the bar for ICLR in its current form. It does not provide sufficient novel theoretical insight, fails to compellingly motivate its contributions, and lacks the clarity and contextual discussion needed to make the results accessible and meaningful to the machine learning community.

**Reviewer Concerns:**

The authors did not provide a rebuttal.

**Reviewer Scores:**

Given the lack of a rebuttal, the reviewers are expected to maintain their original scores.

---

### Decision · Program_Chairs · 2026-01-26

Reject